# Development of a Rapid and Cost-Effective Multiplex PCR Assay for the Simultaneous Identification of Three Commercially Important Sea Squirt Species (*Halocynthia* spp.)

**DOI:** 10.3390/foods14173003

**Published:** 2025-08-27

**Authors:** Kang-Rae Kim, Hye-Jin Kim, In-Chul Bang

**Affiliations:** 1Namdonghae Fishery Research, National Institute of Fishery Science, Namhae 52440, Republic of Korea; krkim89@korea.kr; 2Department of Life Science & Biotechnology, Soonchunhyang University, Asan 336-745, Republic of Korea; n_siho@naver.com

**Keywords:** *Halocynthia roretzi*, *Halocynthia aurantium*, *Halocynthia hilgendorfi ritteri*, multiplex PCR, species identification marker

## Abstract

We developed and validated a rapid, cost-effective multiplex PCR assay targeting mitochondrial cytochrome c oxidase subunit I (COX1) to discriminate three commercially important sea-squirt species, *Halocynthia roretzi*, *H. aurantium* and *H. hilgendorfi ritteri*. Species-specific forward primers were designed from interspecific single-nucleotide polymorphisms within the barcode region and combined with a common reverse primer in a single reaction. Specificity was confirmed in all tested individuals (*n* = 7 per species) without cross-amplification. Sensitivity tests demonstrated consistent amplification down to 0.1 ng of template DNA, matching or surpassing detection limits reported for other food-authentication markers. Because the entire reaction including DNA extraction can be completed within three hours and requires only basic laboratory equipment, the method is well suited for quality control laboratories, border inspections and routine monitoring of processed products. The COX1 multiplex PCR set proposed here provides a reliable tool to enhance traceability, protect consumer choice, and support regulatory enforcement in the sea-squirt supply chain.

## 1. Introduction

Ascidians, commonly known as sea squirts, represent a significant and highly valued component of the global seafood market, particularly within East Asian culinary traditions [1]. Species belonging to the genus *Halocynthia*, such as *Halocynthia roretzi*, *H. aurantium*, and *H. hilgendorfi ritteri*, are commercially important delicacies prized for their unique flavor and texture [2]. However, the very processing methods that prepare these products for market create a critical vulnerability [3]. Sea squirts are typically sold after the removal of their outer tunic, with only the internal muscle tissue being packaged and distributed [3]. This practice, while essential for culinary use, systematically eliminates the key morphological and anatomical features required for traditional, visual species identification [4]. Consequently, once processed, these closely related species become virtually indistinguishable to the naked eye, opening a significant opportunity for economic adulteration and fraud [5,6,7].

Extensive studies have documented high rates of mislabeling in high-value fish products such as tuna, cod, and snapper, where less expensive species are fraudulently substituted for premium ones [5,6,7]. For example, *H. roretzi* producers may sell their products at higher prices when they process them into more expensive varieties at the point of sale, since they are not easily distinguishable visually [8]. Therefore, the development of species identification markers is not simply a scientific development but an economic necessity to protect honest producers, ensure fair trade, and protect consumer rights [9].

In response to the limitations of morphological identification of processed foods, DNA-based methods are the standard [10]. Among these methods, a method called "DNA barcoding" has been proven to be the standard [10,11]. In the animal kingdom, a region of approximately 650 base pairs in the mitochondrial cytochrome c oxidase subunit I (COX1) gene is universally accepted as a standard barcode [10,11]. The COX1 gene contains a highly conserved region that is stably conserved across a wide range of taxa, and universal primers (such as the LCO1490/HCO2198 primers used in this study) that can amplify genes from various species have become the standard [10,11]. This gene has a hypervariable region that accumulates mutations at a rapid rate enough to generate distinct genetic characteristics between closely related species but is generally less variable within a single species [12].

Direct DNA sequencing of the COX1 region provides the most comprehensive data and is invaluable for discovering new species or exploring phylogenetic relationships, but its application to routine food authentication is very limited [13]. Sequencing is inherently complex, time-consuming, and expensive, requiring specialized equipment and significant bioinformatics expertise for data analysis [14,15,16]. This makes it impractical for environments such as quality control laboratories or border inspection stations where large numbers of samples must be processed quickly and inexpensively [17]. An alternative, species-specific PCR, is faster but less efficient because separate reactions are required to test each potential species [5,6]. Despite the commercial importance of *Halocynthia* species and the well-known challenges of seafood authentication, a rapid and cost-effective method for simultaneously identifying *H. roretzi*, *H. aurantium*, and *H. hilgendorfi ritteri* has not been studied. Previous studies have successfully developed multiplex PCR assays for other seafood groups such as tuna and cod, but no tools have been developed and validated specifically for these major sea squirt species [5,6,7,8].

Therefore, the objectives of this study were (1) to reduce the possibility of food fraud by simultaneously, rapidly, and clearly identifying *H. roretzi*, *H. aurantium*, and *H. hilgendorfi ritteri*, (2) to develop a COX1-based multiplex PCR assay, and (3) to validate the developed multiplex marker to provide a practical tool for food certification and regulatory enforcement.

## 2. Materials and Methods

### 2.1. Sample Acquisition and Genomic DNA Extraction

Samples of *Halocynthia roretzi*, *H. aurantium*, and *H. hilgendorfi ritteri* were purchased from seafood vendors in Goseong (38°23′52″ N, 128°29′31″ E) and Tongyeong (34°50′09′′ N, 128°26′05′′ E), South Korea, in November 2020. The muscles inside the epidermis were incised, placed in a 1.5 mL tube containing 99.9% ethanol, and soaked at room temperature (24 °C) for 3 days before being used to extract genomic DNA. For genomic DNA extraction, the samples were washed with triple-distilled water, and genomic DNA was extracted using HiGeneTM Genomic DNA Prep Kit (Biofact, Deajeon, Republic of Korea) according to the manual provided by the manufacturer.

### 2.2. Mitochondrial Cytochrome Coxidase Subunit I (COX1) Amplification

To amplify the COX1 region of *H. roretzi*, *H. aurantium*, and *H. hilgendorfi ritteri*, the universal primers LCO1490 and HCO2198 of Folmer et al. [11] were used. The PCR reaction for amplifying the mitochondrial COX1 region was performed by adding 10 ng of genomic DNA and 10 μM of each marker primer to a 20 μL volume of AccuPower^®^ PCR Premix Kit (Bioneer Inc., Daejeon, Republic of Korea), followed by an initial denaturation reaction at 94 °C for 5 min, followed by 34 cycles of 94 °C for 45 s, annealing at 49 °C for 50 s, and extension at 72 °C for 55 s. The final extension reaction was performed at 72 °C for 7 min. The amplified products were confirmed by electrophoresis on a 1.5% agarose gel.

### 2.3. Species-Specific Primer Design

The base sequences of some regions of COX1 obtained from three species were aligned using the BioEdit program (v7.7.1) [18] to obtain the base sequences. Among the base sequences obtained from three species, a set of primers for amplifiable sea squirts was created by selecting base sequences that are specific between species, excluding base sequences that show intraspecific polymorphism, and Ha, Hr, and Hhr primer sets were created to enable amplification.

### 2.4. Multiplex PCR of Species-Specific Primer Sets for Three Species of Sea Squirts

PCR was performed using each Ha, Hr, and Hhr primer set with 10 ng of three species of genomic DNA as a standard to confirm the amplification of PCR products. 10 μM of each primer was added to a 20 μL AccuPower^®^ Multiplex PCR Premix Kit (Bioneer Inc., Daejeon, Republic of Korea), and the initial denaturation reaction was performed at 94 °C for 5 min, followed by 34 cycles of 94 °C for 30 s, annealing at 55 °C for 30 s, and extension at 72 °C for 30 s. The final extension reaction was performed at 72 °C for 7 min.

### 2.5. Confirmation of Multiplex PCR Amplification According to DNA Concentration

Multiplex PCR reaction was performed using each Ha, Hr, and Hhr primer set with 50, 10, 1, 0.1, and 0.01 ng of three genomic DNAs as the standard to confirm the amplification of PCR products. Multiplex PCR conditions were the same as those in Materials and Methods Section 2.4. Digital quantification of each concentration and species band was performed using ImageJ (v1.54) [19].

## 3. Results and Discussion

### 3.1. Mitochondrial Cytochrome Coxidase Subunit I (COX1) Amplification

For the three species of sea squirts, the LCO1490 and HCO2198 mitochondrial COX1 regions were amplified, and it was confirmed that all PCR amplification bands were amplified 100% in each species (Figure 1). The DNA barcode region used in the animal kingdom should evolve at a sufficiently fast rate to distinguish species so that interspecific variation is large, while intraspecific variation should be low so that species stability can be maintained [20]. The mitochondrial COX1 gene has been proposed as the gene that best meets these criteria and is currently widely recognized as the standard DNA barcode in the animal kingdom [21,22,23]. In this study, we confirmed that the three species of sea squirts were amplified using the commonly used LCO1490 and HCO2198 DNA barcoding primer sets, confirming that the DNA barcoding markers were sufficiently amplified.

### 3.2. Species-Specific Primer Design

Primer sets were designed based on the COX1 sequences of three species of sea squirts (Table 1 and Appendix A). Using specific primer sets for the three species of sea squirts, each amplified band was sequenced to obtain the base sequence, and the WebBLAST (https://blast.ncbi.nlm.nih.gov, accessed 1 March 2020) result of the obtained base sequences on the NCBI database showed that they were 100% identical to the three species of sea squirts (Figure 2). Additionally, specific multiplex primer sets were used to perform amplification on 24 individuals of each species (Appendix A). Accurate species identification is important in a wide range of fields such as ecosystem conservation, food safety management, and quarantine [23]. Traditionally, species identification has been based on anatomical and morphological characteristics, but this approach has limitations in that morphological identification is difficult in the case of professional skills or processed food forms [22,23]. For example, sea squirts are impossible to distinguish morphologically with the naked eye because the epidermis is removed, and the internal flesh is processed. However, genetically distinct species can be accurately and reliably distinguished using species-specific markers. In previous studies on the identification of species consumed as food, species identification markers were developed based on regions with conserved base sequences and large interspecies mutation differences, such as the COX1 and Cytb genes in the mitochondrial region of tuna, Pacific cod, and Skipjack tuna [5,6,7]. In this study, species identification markers were developed for three species of sea squirts mainly used as food, and markers that can be clearly identified by the difference in the size of the amplified band were developed so that they can be useful for species identification in foods or tissues where morphological classification keys do not remain.

### 3.3. Multiplex PCR Set Amplification PCR Band

As a result of performing multiplex PCR with primer sets for three species of sea squirts, the amplification product of *H. hilgendorfi ritteri* was 360 bp, that of *H. aurantium* was 186 bp, and that of *H. roretzi* was 118 bp (Figure 3). Multiplex PCR primer set: the three species could be identified based on the differences in band sizes. The sequencing method directly decodes and provides the base sequence information of the entire COX1 barcode region, and accurate and comprehensive data can be obtained [24]. Sequencing allows for in-depth research, such as identifying unknown species, discovering new genetic mutations, and elucidating the phylogenetic relationships between species [25]. However, DNA sequencing has obvious disadvantages in that the analysis process is complex, expensive, and time-consuming [5]. After purifying the PCR product, performing the sequencing reaction, and analyzing the obtained base sequence data, several follow-up steps are required [18]. It is not suitable for quality control or quarantine sites that require rapid processing of a large number of samples [26]. Considering these technical characteristics, cost-effective and time-efficient marker development is required depending on the purpose of the research application [27,28]. The goal of this study was to quickly and inexpensively identify three commercial sea squirt species when the mitochondrial sequence is already known. No previous studies have developed species-specific markers applicable to sea squirts. Therefore, the multiplex PCR set using the species-specific markers developed in this study can be useful for species identification of three sea squirt species.

### 3.4. Multiplex PCR Set Amplification Sensitivity by DNA Concentration

In this study, we optimized the COX1-based multiplex PCR conditions for the identification of three species and confirmed that reproducible amplification was possible even when the template DNA was diluted to 0.1 ng (Figure 4). This is a sensitivity that is consistent with the detection limit (0.1 ng level) of mitochondrial markers (multiplex PCR) in food matrices reported in previous studies such as Bai et al. [29], Hanapie et al. [30], Kitpipit et al. [31], Sultanae et al. [32], and Boyrusbiantoe et al. [33]. The sea squirt COX1 marker discussed in this study showed stable amplification efficiency at the same limiting concentration despite limited research cases in domestic and foreign food samples, unlike the previously used ND4, cytb, and COI markers (Appendix A) [29,30,31,32,33]. These results imply that rapid and reliable species identification is possible with the COX1-based multiplex PCR protocol in this study when only 0.1 ng or more of DNA is secured from processed aquatic products containing sea squirt components or where cross-contamination is a concern. However, since verification has not been conducted for cases where salting or heating was applied, it is believed that verification of this will be necessary in the future. Therefore, this method satisfies the verification standards required for food hygiene and country of origin labeling systems and can be applied even under conditions of minimal sample amount and low DNA concentration, thereby providing an important marker set for the future identification of sea squirt foods.

## 4. Conclusions

This study presents the first COX1-based multiplex PCR system capable of simultaneously identifying *H. roretzi*, *H. aurantium*, and *H. hilgendorfi ritteri* in a single tube. The primer set produced species-specific bands that remained clearly distinguishable even when template DNA was diluted to 0.1 ng, demonstrating high analytical sensitivity suitable for highly processed or mixed seafood matrices. The assay requires no expensive instrumentation, shortens analysis time, and lowers costs compared with sequencing-based approaches, making it a practical alternative for routine authenticity testing.

## Figures and Tables

**Figure 1 foods-14-03003-f001:**
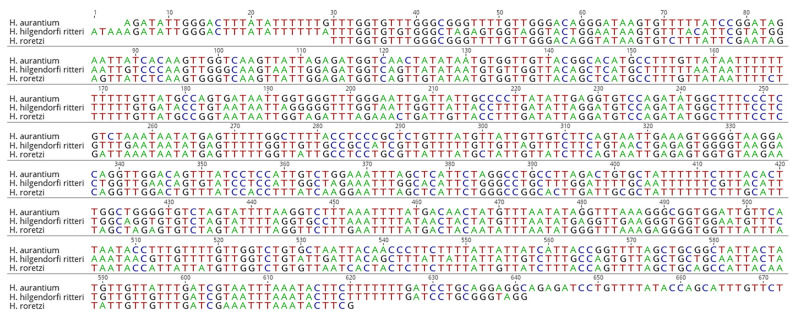
Results of amplification of a COX1 for three species of sea squirts using the LCO1490 and HCO2198 primer sets and base sequence alignment.

**Figure 2 foods-14-03003-f002:**
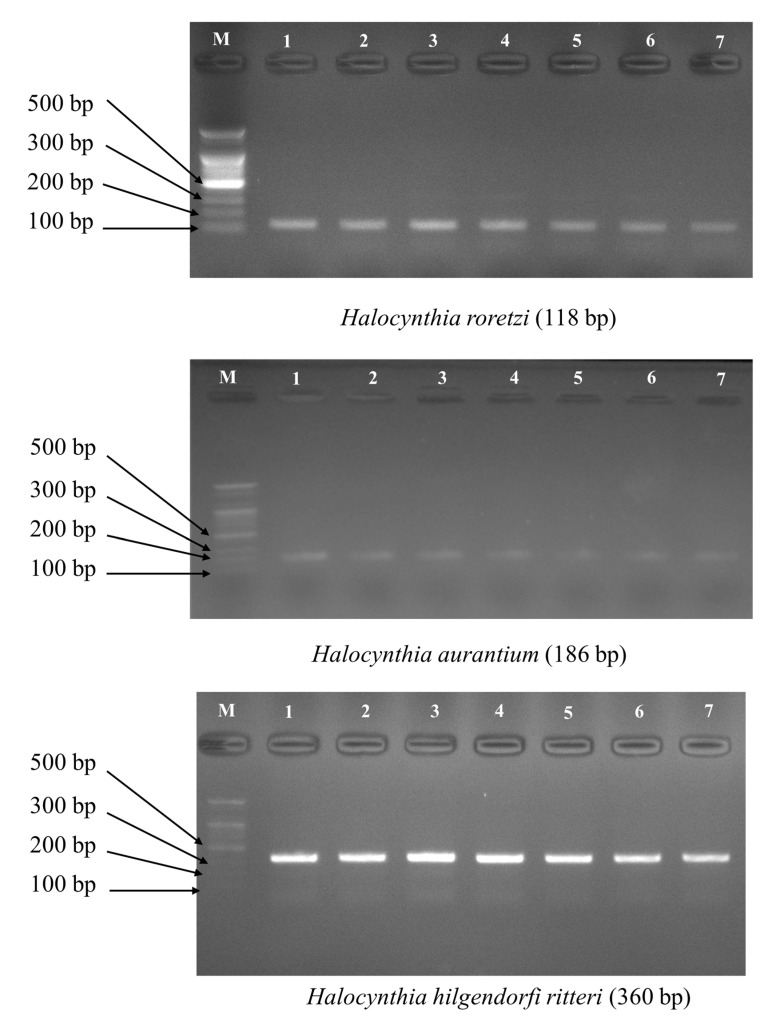
Species-specific PCR primer amplification results for single species of three species of sea squirts (Primer set of *H. roretzi* PCR band: Hr_CO1_F & Pyuridae_CO1_R; Primer set of *H. aurantium* PCR band: Ha_CO1_F & Pyuridae_CO1_R; Primer set of *H. hilgendorfi ritteri* PCR band: Hhr_CO1_F & Pyuridae_CO1_R).

**Figure 3 foods-14-03003-f003:**
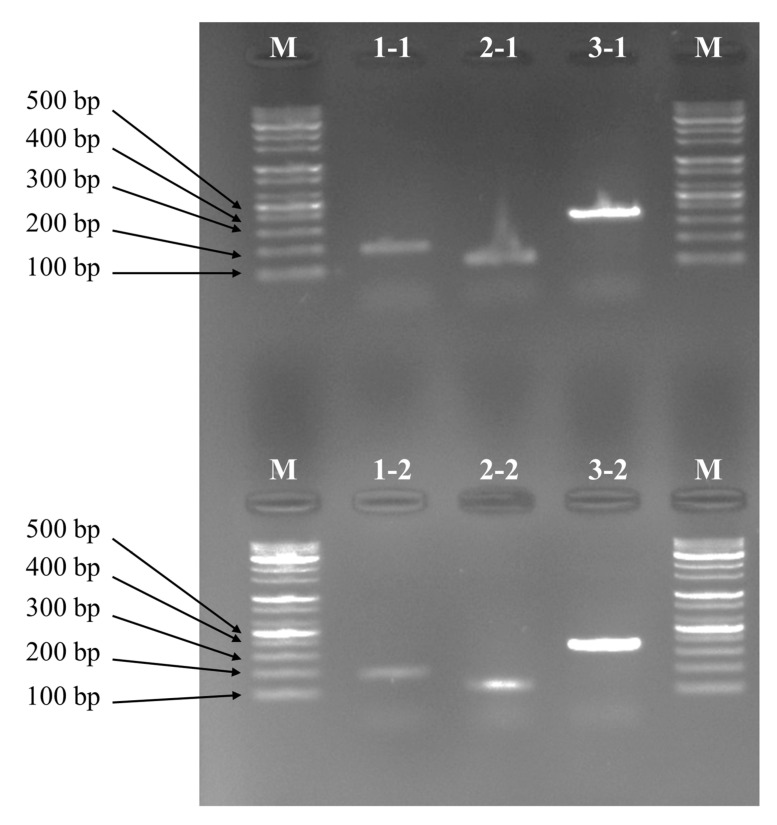
Results of amplification using a multiplex PCR primer set. M: Represents DNA ladder. 1-1: PCR band of individual 1 of *H. aurantium* (186 bp), 2-1: PCR band of individual 1 of *H. roretzi* (118 bp), 3-1: PCR band of individual 1 of *H. hilgendorfi ritteri* (360 bp). 1-2: PCR band of individual 2 of *H. aurantium* (186 bp), 2-2: PCR band of individual 2 of *H. roretzi* (118 bp), 3-2: PCR band of individual 2 of *H. hilgendorfi ritteri* (360 bp).

**Figure 4 foods-14-03003-f004:**
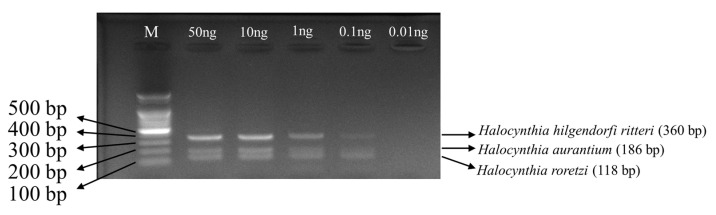
PCR bands by concentration according to DNA pooling of three species of sea squirts using a multiplex PCR set. PCR amplification was performed by mixing the DNA of three species of sea squirts and amplifying them at different concentrations using a multiplex primer set. No band amplification at 0.01 ng.

**Table 1 foods-14-03003-t001:** Primer set design information based on amplified COX1 region for three species of sea squirts.

**Primer Name**	**Species**	**Sequence(5′-3′)**	**Primer Direction**	**Product Size (bp)**
Hhr_CO1_F	*H. hilgendorfi ritteri*	TTGGTGTGTGGGCTAGAGTG	Forward	360
Ha_CO1_F	*H. aurantium*	GATTATTGCCCCTTATATTGAGG	Forward	184
Hr_CO1_F	*H. roretzi*	TGGTTATTGCCTCCTGCGT	Forward	118
Pyuridae_CO1_R	-	CMGGCCYAGAATGWGCYAA	Reverse	-

## Data Availability

The original contributions presented in the study are included in the article and Appendix A, further inquiries can be directed to the corresponding author.

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
