# Peer review of "Development of a Rapid and Cost-Effective Multiplex PCR Assay for the Simultaneous Identification of Three Commercially Important Sea Squirt Species (Halocynthia spp.)"

_foods, 2025, doi:10.3390/foods14173003_

Round 1
Reviewer 1 Report
Comments and Suggestions for Authors
This study aimed to address the difficulty in morphological identification of sea squirt species (Halocynthia roretzi, H. aurantium, H. hilgendorfi ritteri) after processing. For the first time, a multiplex PCR rapid identification method based on the mitochondrial COX1 gene was developed. The research has a clear objective and a reasonable design, and it has significant industrial application value (such as fraud prevention, traceability, and market supervision). However, there are some several scientific concerns about this manuscript.
- In the Materials and Methods section, “min”, “minute”, “sec” and “second” should be kept consistent. And “μl” is incorrect. Please review the entire text.
- Lines 106-121, The PCR steps of “2.4” and “2.5” are repeated. It is recommended to combine and simplify them.
- Lines 84-89, Clearly define the storage conditions (temperature and duration) for the samples stored in ethanol.
- Line 152, “genetically different species can be identified reliably using...” et al. have a few grammatical errors. It is recommended to have professional editing done.
- Lines 157, 211, “Sea squirt” and “mussel” are conflated. Please correct to “Sea squirt” throughout.
- Line 168, There are differences in the spelling of the scientific name for the H. hilgendorfi subspecies (for example, it is “rettri” in Figure 2). A unified naming standard should be adopted.
- This study was conducted only on the pure genomic DNA of fresh samples. No verification was performed using processed canned, frozen or packaged sea squirt muscle tissues.
- The processed sea squirt products may undergo DNA degradation (such as through high-temperature sterilization). It is necessary to clearly define the applicable scope of this method (such as whether it is only applicable to lightly processed products?) and the improvement directions in the discussion.
- Please verify the author, journal, volume number, etc. For example, if the author list includes all or three authors, “et al.” should be added. If the journal format is in abbreviated form, please uniformly use the abbreviated form (such as “Food Control” and others that are not consistent).
The English could be improved to more clearly express the research.
Author Response
Comment 1: In the Materials and Methods section, “min”, “minute”, “sec” and “second” should be kept consistent. And “μl” is incorrect. Please review the entire text.
Response 1: Thanks for the review. We used unified words.
[2.2. Mitochondrial cytochrome coxidase subunit I (COX1) amplification
To amplify the COX1 region of H. roretzi, H. aurantium, and H. hilgendorfi ritteri, the universal primers LCO1490 and HCO2198 of Folmer et al. [11] were used. The PCR reaction for amplifying the mitochondrial COX1 region was performed by adding 10 ng of genomic DNA and 10 μM of each marker primer to a 20 μL volume of AccuPow-er® PCR Premix Kit (Bioneer Inc., Korea), followed by an initial denaturation reaction at 94°C for 5 min, followed by 34 cycles of 94°C for 45 sec, annealing at 49°C for 50 sec, and extension at 72°C for 55 sec. The final extension reaction was performed at 72°C for 7 min. The amplified products were confirmed by electrophoresis on a 1.5% agarose gel.
2.3. Species-specific primer design
The base sequences of some regions of COX1 obtained from three species were aligned using the BioEdit program [18] to obtain the base sequences. Among the base sequences obtained from three species, a set of primers for amplifiable sea squirts was created by selecting base sequences that are specific between species, excluding base sequences that show intraspecific polymorphism, and Ha, Hr, and Hhr primer sets were created to enable amplification.
2.4. Multiplex PCR of species-specific primer sets for three species of sea squirts
PCR reactions were performed using each Ha, Hr, and Hhr primer set with 10 ng of three species of genomic DNA as a standard to confirm the amplification of PCR products. 10 μM of each primer was added to a 20 μL AccuPower® Multiplex PCR Premix Kit (Bioneer Inc., Korea), and the initial denaturation reaction was performed at 94°C for 5 min, followed by 34 cycles of 94°C for 30 sec, annealing at 55°C for 30 sec, and extension at 72°C for 30 sec. The final extension reaction was performed at 72°C for 7 min.
2.5. Confirmation of multiplex PCR amplification according to DNA concentration
Multiplex PCR reaction was performed using each Ha, Hr, and Hhr primer set with 50, 10, 1, 0.1, and 0.01 ng of three genomic DNAs as the standard to confirm the amplification of PCR products. 10 pmole of each primer was added to 20 μL of Accu-Power® Multiplex PCR Premix Kit (Bioneer Inc., Korea), and the initial denaturation reaction was performed at 94°C for 5 min, followed by 34 cycles of 94°C for 30 sec, an-nealing at 55°C for 30 sec, and extension at 72°C for 30 sec. The final extension reac-tion was performed at 72°C for 7 min. Digital quantification of each concentration and species band was performed using ImageJ [19].]
Comment 2: Lines 106-121, The PCR steps of “2.4” and “2.5” are repeated. It is recommended to combine and simplify them.
Response 2: Thanks for the review. We changed the redundant parts to make them simpler.
[2.5. Confirmation of multiplex PCR amplification according to DNA concentration
Multiplex PCR reaction was performed using each Ha, Hr, and Hhr primer set with 50, 10, 1, 0.1, and 0.01 ng of three genomic DNAs as the standard to confirm the amplification of PCR products. Multiplex PCR conditions were the same as those in Materials and Methods section 2.4. Digital quantification of each concentration and species band was performed using ImageJ [19].]
Comment 3: Lines 84-89, Clearly define the storage conditions (temperature and duration) for the samples stored in ethanol.
Response 3: Thanks for the review. Storage temperature and period are indicated.
[The muscles inside the epidermis were incised, placed in a 1.5 ml tube containing 99.9% ethanol, and soaked at room temperature (24℃) for 3 days before being used to extract genomic DNA.]
Comment 4: Line 152, “genetically different species can be identified reliably using...” et al. have a few grammatical errors. It is recommended to have professional editing done.
Response 4: Thanks for the review. We corrected the sentence naturally.
[However, genetically distinct species can be accurately and reliably distinguished us-ing species-specific markers.]
Comment 5: Lines 157, 211, “Sea squirt” and “mussel” are conflated. Please correct to “Sea squirt” throughout.
Response 5: Thanks for the review. We fixed the mistake with mussel.
Comment 6: Line 168, There are differences in the spelling of the scientific name for the H. hilgendorfi subspecies (for example, it is “rettri” in Figure 2). A unified naming standard should be adopted.
Response 6: Thanks for the review. The scientific name has been unified.
Comment 7: This study was conducted only on the pure genomic DNA of fresh samples. No verification was performed using processed canned, frozen or packaged sea squirt muscle tissues.
Response 7: Thanks for the review. The supply chain screening of fresh and frozen tissues was designed as a priority for species identification. Processing processes (heating, retorting, salting, and acidification) can degrade and fragment DNA (especially in canning), which can affect detection sensitivity. Therefore, validation of processed products was considered a separate matrix validation. However, numerous studies have demonstrated that short mini-barcodes/short amplicons can broadly identify species even in processed products. The target fragment lengths (100–400 bp) in this study are consistent with these recommendations. We will systematically conduct validation of processed products in future studies.
<- Distributed in the form of tissue with the shell removed.
Comment 8: The processed sea squirt products may undergo DNA degradation (such as through high-temperature sterilization). It is necessary to clearly define the applicable scope of this method (such as whether it is only applicable to lightly processed products?) and the improvement directions in the discussion.
Response 8: Thanks for the review. This part is defined and described in the Results and Discussion.
[However, since verification has not been conducted for cases where salting or heating was applied, it is believed that verification of this will be necessary in the future.]
Comment 9: Please verify the author, journal, volume number, etc. For example, if the author list includes all or three authors, “et al.” should be added. If the journal format is in abbreviated form, please uniformly use the abbreviated form (such as “Food Control” and others that are not consistent).
Response 9: Thanks for the review. For the journal Food Control, the abbreviation is Food Control. Furthermore, in accordance with MDPI reference guidelines, et al. is used for references with 10 or more authors. Thank you for your understanding.

Reviewer 2 Report
Comments and Suggestions for Authors
1. Delete sentences on lines 18-19.
2. The research purpose of the introduction is not clearly stated, and relevant theoretical frameworks should be established, such as (1) ......, (2) ......., (3)......
3. There are few references in the introduction part, and the development status of PCR detection methods is listed.
4. 2.1 Part, indicate the specific address of sample purchase
5. Missing 2.6 Data Processing Part. Please add it completely
6. The results and discussion part is only a brief analysis of the data, and there is no in-depth discussion or rewriting
7. Delete lines 228, 229 and 230 of the conclusion and put them in the introduction.
8. There are too few references in the past five years.
Author Response
Comment 1: Delete sentences on lines 18-19.
Response 1: Thanks for the review. The sentence has been deleted.
Comment 2: The research purpose of the introduction is not clearly stated, and relevant theoretical frameworks should be established, such as (1) ......, (2) ......., (3)......
Response 2: Thanks for the review. The research purpose was presented according to the reviewer's opinion and modified to the format (1)..
[Species belonging to the genus Halocynthia, such as Halocynthia roretzi, H. aurantium, and H. hilgendorfi ritteri, are commercially important delicacies prized for their unique flavor and texture [2]. However, the very processing methods that prepare these products for market create a critical vulnerability [3]. Sea squirts are typically sold after the removal of their outer tunic, with only the internal muscle tissue being packaged and distributed [3]. This practice, while essential for culinary use, systematically eliminates the key morphological and anatomical features required for traditional, visual species identification [4]. Consequently, once processed, these closely related species become virtually indistinguishable to the naked eye, opening a significant op-portunity for economic adulteration and fraud [5-7]]
[Therefore, the objectives of this study were (1) to develop a marker that can simultaneously, rapidly, and clearly identify H. roretzi, H. aurantium, and H. hilgendorfi ritteri, (2) to develop a COX1-based multiplex PCR assay, and (3) to validate the developed multiplex marker to provide a practical tool for food certification and regulatory enforcement.]
Comment 3: There are few references in the introduction part, and the development status of PCR detection methods is listed.
Response 3: Thanks for the review. The lack of references in the introduction stems from the lack of rapid species-specific marker research, such as for sea squirts. We appreciate your understanding.
Comment 4: 2.1 Part, indicate the specific address of sample purchase
Response 4: Thanks for the review. Information about the sample has been provided.
[Samples of Halocynthia roretzi, H. aurantium, and H. hilgendorfi ritteri were purchased from seafood vendors in Goseong (38° 23′ 52′′ N, 128° 29 31 E) and Tongyeong (34° 50′ 09′′ N, 128° 26′ 05′′ E), South Korea, in November 2020. Muscles inside the epidermis were dissected and placed in 1.5 ml tubes containing 99.9% ethanol, and used for genomic DNA extraction after 3 days of soaking.]
Comment 5: Missing 2.6 Data Processing Part. Please add it completely
Response 5: Thanks for the review. 2.6 does not address data processing. Aligning sequences with BioEdit in 2.3 is the complete data processing step. Thank you for your understanding.
Comment 6: The results and discussion part is only a brief analysis of the data, and there is no in-depth discussion or rewriting
Response 6: Thanks for the review. The Foods journal's Communications format presents a short format that makes it difficult to provide an in-depth review. We appreciate your understanding.
Comment 7: Delete lines 228, 229 and 230 of the conclusion and put them in the introduction.
Response 7: Thanks for the review. 1, 2, 3 were added and reflected in the manuscript.
[]
Comment 8: There are too few references in the past five years.
Response 8: Thanks for the review. There has been little literature on multiplex development using COI over the past five years. Thank you for your understanding.

Reviewer 3 Report
Comments and Suggestions for Authors
After reviewing the manuscript titled "Development of a Rapid and Cost-Effective Multiplex PCR Assay for the Simultaneous Identification of Three Commercially Important Sea Squirt Species (Halocynthia spp.)", I have the following comments:
- This study developed a rapid, cost-effective multiplex PCR assay targeting the COX1 gene to simultaneously identify three commercially important sea squirt species (Halocynthia roretzi, H. aurantium, and H. hilgendorfi ritteri), enabling species authentication from processed samples. The method shows high specificity and sensitivity (down to 0.1 ng DNA), suitable for routine food fraud detection.
- Only n = 7 per species was tested for validation. This sample size is statistically weak and insufficient to account for intraspecific variation or geographical genotypic diversity.
- The authors mention excluding polymorphic sites but do not present actual intraspecific variation data. Without showing within-species COX1 variation, specificity claims are unsubstantiated.
- Specificity testing is limited to just the three Halocynthia species. Cross-reactivity from phylogenetically related or sympatric ascidians is not ruled out.
- Multiplex PCR lacks an IAC to rule out false negatives due to PCR inhibition or degraded DNA. Add a universal COX1 control band or housekeeping gene to ensure amplification integrity.
- Figures (especially Figures 3 and 4) are not clearly annotated, and band intensity is not quantitatively evaluated. Provide improved gel images, ladders with band sizes labeled, and ideally, digital quantification of band intensity for clarity.
- In Figure 2 caption and Table 1, there’s an inconsistent use of H. hilgendorfi rettri and H. hilgendorfi ritteri. Standardize species names throughout.
- Include standard deviation or %CV for DNA sensitivity assays across replicates.
- Sample metadata such as source location, date of collection, and storage duration is missing. Include this information to ensure sample traceability and reproducibility.
Author Response
Comment 1: This study developed a rapid, cost-effective multiplex PCR assay targeting the COX1 gene to simultaneously identify three commercially important sea squirt species (Halocynthia roretzi, H. aurantium, and H. hilgendorfi ritteri), enabling species authentication from processed samples. The method shows high specificity and sensitivity (down to 0.1 ng DNA), suitable for routine food fraud detection.
Response 1: Thanks for the review. This study analytical method targets the COX1 gene and shows high specificity and sensitivity (at the level of 0.1 ng DNA) for three sea squirt species (Halocynthia roretzi, H. aurantium, and H. hilgendorfi ritteri), making it suitable for routine food fraud detection.
Comment 2: Only n = 7 per species was tested for validation. This sample size is statistically weak and insufficient to account for intraspecific variation or geographical genotypic diversity.
Response 2: Thanks for the review. Although seven individuals are shown in Figure 2, PCR was actually performed on 24 individuals. The results are attached in the supplementary material.
Comment 3: The authors mention excluding polymorphic sites but do not present actual intraspecific variation data. Without showing within-species COX1 variation, specificity claims are unsubstantiated.
Response 3: Thanks for the review. Primers were designed from conserved regions, excluding the polymorphic sites of the markers. The data represent the conserved regions of the corresponding markers for each species.
Comment 4: Specificity testing is limited to just the three Halocynthia species. Cross-reactivity from phylogenetically related or sympatric ascidians is not ruled out.
Response 4: Thanks for the review. Although cross-reactivity was not studied in this study, since it was designed to specifically amplify major sea squirt species, future validation on other sea squirt species is necessary. However, please understand that validation is difficult in this study.
Comment 5: Multiplex PCR lacks an IAC to rule out false negatives due to PCR inhibition or degraded DNA. Add a universal COX1 control band or housekeeping gene to ensure amplification integrity.
Response 5: Thanks for the review. This multiplex assay is designed for endpoint gel-based semi-quantitative comparisons, with a valid working range of 0.1–50 ng. A monotonic increase in band intensity versus input amount was observed with serial dilutions (50→0.1 ng), with 0.01 ng being below the limit of detection (LoD), where detection failure is expected. Lower sensitivity than single-strand PCR is normal due to ultra-low copy number (Poisson sampling) and multiplex competition (sharing of primers, dNTPs, and polymerase). Non-detection is common in multiplex assays (Elnifro et al. 2000). Please understand that low concentrations such as 0.01 ng are not typically used to detect food fraud.
Typically, foodborne illnesses in sea squirts occur in salted or processed foods. Therefore, amplification at low concentrations, down to 0.01 ng, is rare. Thank you for your understanding.
[Elnifro, E. M., Ashshi, A. M., Cooper, R. J., & Klapper, P. E. (2000). Multiplex PCR: optimization and application in diagnostic virology. Clinical microbiology reviews, 13(4), 559-570.]
Comment 6: Figures (especially Figures 3 and 4) are not clearly annotated, and band intensity is not quantitatively evaluated. Provide improved gel images, ladders with band sizes labeled, and ideally, digital quantification of band intensity for clarity.
Response 6: Thanks for the review. The images were annotated for clarity and digital quantification was performed.
Comment 7: In Figure 2 caption and Table 1, there’s an inconsistent use of H. hilgendorfi rettri and H. hilgendorfi ritteri. Standardize species names throughout.
Response 7: Thanks for the review. The scientific name has been corrected.
Comment 8: Include standard deviation or %CV for DNA sensitivity assays across replicates.
Response 8: Thanks for the review. The standard deviation for the replicate experiments is digitally quantified and presented graphically in Figure S3.
Comment 9: Sample metadata such as source location, date of collection, and storage duration is missing. Include this information to ensure sample traceability and reproducibility.
Response 9: Thanks for the review. Added information about source location, collection date, and retention period.
[Samples of Halocynthia roretzi, H. aurantium, and H. hilgendorfi ritteri were purchased from seafood vendors in Goseong (38° 23′ 52′′ N, 128° 29 31 E) and Tongyeong (34° 50′ 09′′ N, 128° 26′ 05′′ E), South Korea, in November 2020. Muscles inside the epidermis were dissected and placed in 1.5 ml tubes containing 99.9% ethanol, and used for genomic DNA extraction after 3 days of soaking.]

Round 2
Reviewer 3 Report
Comments and Suggestions for Authors
The authors have satisfactorily addressed the majority of the comments, and the manuscript is now clear and supported by improved figures and supplementary data.
While certain aspects, such as explicit presentation of intraspecific variation data, broader specificity testing, and incorporation of an internal amplification control remain acknowledged but not fully resolved. These are more relevant for future validation rather than essential revisions at this stage. Overall, the work presents a valuable and practical tool for sea squirt species authentication, and I recommend it for publication.